# More Stages Decrease Dissipation in Irreversible Step Processes

**DOI:** 10.3390/e25030539

**Published:** 2023-03-21

**Authors:** Peter Salamon, Bjarne Andresen, James Nulton, Ty N. F. Roach, Forest Rohwer

**Affiliations:** 1Department of Mathematics, San Diego State University, San Diego, CA 92182, USA; jnulton@sdsu.edu; 2Niels Bohr Institute, University of Copenhagen, Blegdamsvej 17, DK-2100 Copenhagen, Denmark; 3Department of Biology, San Diego State University, San Diego, CA 92182, USA; smokinroachjr@gmail.com (T.N.F.R.); frohwer@sdsu.edu (F.R.)

**Keywords:** minimizing dissipation, relaxation, step processes, sequential processes, thermodynamic length

## Abstract

The dissipation in an irreversible step process is reduced when the number of steps is increased in any refinement of the steps in the process. This is a consequence of the ladder theorem, which states that, for any irreversible process proceeding by a sequence of relaxations, dividing any relaxation step into two will result in a new sequence that is more efficient than the original one. This results in a more-steps-the-better rule, even when the new sequence of steps is not reoptimized. This superiority of many steps is well established empirically in, e.g., insulation and separation applications. In particular, the fact that the division of any step into two steps improves the overall efficiency has interesting implications for biological evolution and emphasizes thermodynamic length as a central measure for dissipation.

## 1. Introduction

Finite-time thermodynamics (FTT) was originally developed to make the losses of a process out of equilibrium explicitly visible, “the cost of haste” [1,2]. Subsequently, these tools were generalized into control thermodynamics [3], i.e.,: To which degree can one guide a thermodynamic process toward a desired result, what is the minimal cost, and how can one achieve it? One of the important components in the theory is thermodynamic length. It has been used to bound the dissipations of an irreversible process, originally in the continuous case [1], shortly after in the discrete case [4]. In both cases, the dissipation is limited by
(1)$$\Delta S^{u}\geq \frac{L^{2}}{2K}$$
where we have chosen the entropy production of the process, $\Delta S^{u}$, as the measure of dissipation. *L* is the thermodynamic length of the process, and *K* is the number of discrete steps when we consider the process taking place as a sequence of small equilibrations, i.e., all endpoints of the equilibrations are in equilibrium with the reservoir. Actually, we can almost always describe dissipation as due to appropriately constrained relaxations. We think of these points of equilibrium with the sequence of reservoirs as the rungs of the ladder and, thus, the equilibrations as the spaces between neighboring rungs of the ladder. We will return more specifically to this analogy below.

From the functional form of Equation (Equation 1), we observe two important points. The first and most-obvious is that we can reduce the minimum dissipation by increasing the number of steps (*K*). This is our focus in the present paper. The thermodynamic length *L* is fixed by the endpoints of the reaction and by the kind of process we are considering (e.g., isothermal, isobaric, adiabatic) and, thus, is not part of the optimization that leads to Inequality (Equation 1). The second observation is that since the minimum dissipation is quadratic in the length $L_{i}$ of each step, while the sum of the $L_{i}$ is fixed, we will achieve the least entropy production if all the *K* steps are of equal thermodynamic length, L/K. This feature, while present and important for some of our examples, was not our primary focus here. Rather, the feature we focused on is that *increasing the number of steps K by adding an additional step between two existing steps will always decrease dissipation*. We prove this ladder theorem [5] in Appendix A below. It follows from this theorem that any refinement of a step process has lower dissipation, where refinement is meant in the sense of refinement for a partition of an interval [6], i.e., keeping all the previous equilibrium points, but adding more of them. In this sense, we state the more-steps-the-better rule: for any step process, any refinement of the process has lower dissipation.

## 2. A Horse–Carrot Process

Consider a thermodynamic system brought along a particular sequence of equilibrium states via a sequence of relaxations. This is the classic quasistatic process that, in the limit of infinitely many steps, allows the dissipation to go to zero. At each point along the process, the system (the horse) relaxes toward a bath (the carrot), which is in a state just ahead of the current state of the system. In the quasistatic limit, the baths are only infinitesimally ahead and the time of the process goes to infinity. This situation is depicted in Figure 1. To a nearly reversibly evolving system, time evolution always looks like a relaxation. The horse–carrot theorem Equation (Equation 1) [1,4], proven in Appendix A, shows that minimizing the dissipation of a process can be accomplished by either increasing the number of steps *K* (which must all be small) or, if possible, by changing the process along the way so as to shorten the thermodynamic length *L* between the initial and final states.

The minimum entropy production solution is achievable and has been used in the design of an energy efficient distillation column [7,8]. The theorem is also relevant to biology. Frederiksen and Andresen [9,10] calculated the entropy production associated with the cytochrome chain in human mitochondria in which the chemical potential of an electron is lowered by sequential stages to extract work and produce, ultimately, ATP molecules. They demonstrated that the spacing of electron carrier molecules in actual mitochondria serves to produce close to the minimum entropy for a *K*-step process.

## 3. Adding an Intermediate Step

The horse–carrot theorem Equation (Equation 1) was based on an optimal distribution of all the steps in the process. However, even if the steps were not optimally distributed and one adds an additional step anywhere in this ladder (see Figure 2), it will result in a reduction of the dissipation.

Let us illustrate this with a simple process with only a single degree of freedom, volume *V* with the conjugate variable pressure *p* [11]. To keep to this level of simplicity, we assumed that the entire process is carried out isothermally at temperature *T*. Take one of the steps in the sequence where the internal pressure varies from $p_{0}$ to $p_{bath}\left(p_{0}>p_{bath}\right)$. The step is a relaxation process in which the gas, previously in equilibrium with a reservoir at temperature *T* and pressure $p_{0}$, is placed in contact with a reservoir at the next state, i.e., still at temperature *T*, but now with pressure $p_{bath}$. The relaxation involves the volume of the gas increasing from the equilibrium volume $V(p_{0})$ to the next equilibrium volume $V(p_{bath})$. To measure the dissipated work, we subtract the actual work from the maximum possible work that could have been collected from such an expansion.

This maximum possible work is the reversible work and can be obtained while the state of the gas follows the equilibrium isotherm drawn in red in Figure 3: $$W_{reversible}=\int_{V(p_{0})}^{V(p_{bath})}p\left(V\right)dV$$
The actual work performed follows rather the heavy blue line in Figure 3. The external pressure is instantaneously switched to the constant $p_{bath}$, and so, the work performed on the reservoir is
$$W_{actual}=\int_{V(p_{0})}^{V(p_{bath})}p_{bath}dV=p_{bath}\Delta V$$
The dissipation for such a process can be found by subtracting the work performed on the bath, $\int p_{bath}dV$, from the reversible work, ∫pdV, that could have been captured. This gives
(2)$$T\Delta S_{1}^{u}=\int_{V(p_{0})}^{V(p_{bath})}\left(p-p_{bath}\right)dV$$
for the dissipated work converted to heat at *T* and producing entropy $\Delta S_{1}^{u}$. Writing the dissipation as the integral of the difference has some intuitive appeal as it is suggestive of an endoreversible realization of the process, but the final result, $\Delta S_{1}^{u}$, is independent of any details of the expansion. Performing the relaxation in a single step to $p_{bath}$ (blue line in Figure 3) dissipates the work corresponding to the area between the red equilibrium isotherm and the actual path, i.e., the orange + yellow + dark-green areas.

We next introduce an additional intermediate reservoir $p_{inter}$ between $p_{0}$ and $p_{bath}$. To break up the process, we start by relaxing in contact with this intermediate reservoir until some volume $V_{switch}$, with $V_{switch}<V\left(p_{inter}\right)$, at which point we again switch the contact to the final reservoir at $p_{bath}$. Note that the system does not reach equilibrium with the bath at this new switching point, unlike the other points $p_{0}$ and $p_{bath}$. It now follows the dashed heavy blue line, which coincides with the solid heavy blue line for the subsequent relaxation in contact with $p_{bath}$ from $V_{switch}$ to equilibrium at $V_{bath}$. The resulting dissipation becomes
(3)$$T\Delta S_{2}^{u}=\int_{V_{0}}^{V_{switch}}\left(p-p_{inter}\right)dV+\int_{V_{switch}}^{V_{bath}}\left(p-p_{bath}\right)dV.$$
In Figure 3, the areas of dissipation corresponding to these two integrals are shown as the orange and yellow regions, respectively. Adding to this expression the positive quantity: (4)$$\int_{V_{0}}^{V_{switch}}\left(p_{inter}-p_{bath}\right)dV>0$$
(area of the dark-green rectangle in Figure 3), we arrive at
(5)$$T\Delta S_{2}^{u}\leq T\Delta S_{2}^{u}+\int_{V_{0}}^{V_{switch}}\left(p_{inter}-p_{bath}\right)dV.$$
Substituting from (3), we obtain
(6)$$T\Delta S_{2}^{u}\leq \int_{V_{0}}^{V_{switch}}\left(p-p_{inter}\right)dV+\int_{V_{switch}}^{V_{bath}}\left(p-p_{bath}\right)dV+\int_{V_{0}}^{V_{switch}}\left(p_{inter}-p_{bath}\right)dV.$$
The first and third integrals both run from $V_{0}$ to $V_{switch}$ and can be combined by adding the integrands to give
$$\begin{array}{ccc} T\Delta S_{2}^{u} & \leq & \int_{V_{0}}^{V_{switch}}\left[\left(p-p_{inter}\right)+\left(p_{inter}-p_{bath}\right)\right]\;dV+\int_{V_{switch}}^{V_{bath}}\left(p-p_{bath}\right)\;dV\\ \;\; & = & \int_{V_{0}}^{V_{switch}}\left(p-p_{bath}\right)\;dV+\int_{V_{switch}}^{V_{bath}}\left(p-p_{bath}\right)\;dV. \end{array}$$
Since the integrands in the remaining two integrals are identical, we can combine the intervals of integration to give
(7)$$T\Delta S_{2}^{u}\leq \int_{V_{0}}^{V_{bath}}\left(p-p_{bath}\right)dV=T\Delta S_{1}^{u}.$$
It is clear from the figure that the orange + yellow region of dissipation is smaller than the original dissipative region by exactly the dark green rectangle. This reduced dissipation $\Delta S_{1}^{u}-\Delta S_{2}^{u}$ is the benefit of the extra step at $p_{inter}$.

It is clear that, *wherever* we put an intermediate step, it will reduce the dissipation. If we put it optimally, it will be even better. Note that the integration limit $V_{switch}$ (the switching point) does not even need to be at the new reservoir pressure $p_{inter}$, i.e., full equilibration with the new reservoir at $p_{inter}$ is not necessary.

Our primary interest in the present paper was the straightforward observation that adding a step anywhere in any *K*-step process will reduce its entropy production. Let *X* be the set of independent extensive variables of the system. The general case follows by considering breaking a step ΔX into two sub-steps $\Delta X_{1},\Delta X_{2}$, with $\Delta X=\Delta X_{1}+\Delta X_{2}$. For small steps, we can use the equality between entropy production and squared thermodynamic distance in a small relaxation (see Appendix A or [4]):$$\begin{array}{ccc} \Delta S^{u} & = & \frac{1}{2}{\left||\Delta X|\right|}^{2}=\frac{1}{2}||\Delta X_{1}+\Delta X_{2}{\left|\right|}^{2}=\frac{1}{2}\left(\Delta X_{1}+\Delta X_{2}\right)\cdot_{g}\left(\Delta X_{1}+\Delta X_{2}\right)\\ & = & \frac{1}{2}\left(||\Delta X_{1}{\left|\right|}^{2}+||\Delta X_{2}{\left|\right|}^{2}+2\Delta X_{1}\cdot_{g}\Delta X_{2}\right)\\ & = & \Delta S^{u}{{|}_{1}+\Delta S^{u}|}_{2}+\Delta X_{1}\cdot_{g}\Delta X_{2} \end{array}$$
We see that $\Delta S^{u}>\Delta S^{u}{{|}_{1}+\Delta S^{u}|}_{2}$ provided only that the dot product $\Delta X_{1}\cdot_{g}\Delta X_{2}$ is positive, i.e., provided that $\Delta X_{1}$ and $\Delta X_{2}$ make an angle less than 90 deg, so that they move more or less in the same direction. In this flat geometry (approximating the second derivative of S by a constant matrix), the answer to “which states are between two given states” is a sphere, centered on the midpoint between the states. Any state inside this sphere will reduce the total entropy production. While the sphere applies exactly only for small steps, we remark that most reasonable notions of in-between give large regions of validity even for large steps. We then define a **refinement** of a K-step process to be a step process that iterates the addition of one additional step between two existing steps. The new point added must result in two sub-steps that go in approximately the same direction, i.e., have a positive inner product. We then find that the refinement of a $K_{1}$ step process to a $K_{2}$ step process, with $K_{2}>K_{1}$, decreases the dissipation $\Delta S^{u}\left(K_{2}\right)<\Delta S^{u}\left(K_{1}\right)$. We refer to this result as the *ladder theorem*.

Achieving the bound in Equation (Equation 1) involves resizing all steps optimally when adding a new one. However, the ladder theorem is less restrictive. It does not require a full reoptimization of all the steps. It applies even when a single new step is added between existing steps without changing the rest of the chain. Admittedly, this latter step addition in most cases will not improve the chain efficiency quite as much as a full reoptimization of all the steps, but it will improve it.

Note that one consequence of optimally adding an additional step is that all the steps become smaller. If the work extraction process is continuous, that is of no importance, but if work is extracted in “parcels”, e.g., in ATP molecules, as in most cells, some steps may become less than the size of this “parcel” and, thus, can no longer be produced. Within such constraints, increasing the number of steps is no longer feasible. A second consequence of diminishing the losses in the process is that it will physically run more slowly; there is less drive (loss of free energy) to push it forward. This touches on the general issue of optimizing for maximum efficiency (minimum losses) or for maximum power (maximum rate) [12], where the explicit tradeoff between efficiency and power leads to a set of Pareto optima, which optimize some combination of dissipation and rate under the given conditions of the system [13].

One troubling aspect in the derivations above is the proviso that all the steps in the process be small. It turns out that this assumption is not needed. While this manuscript was out for review, we managed to prove our conjecture globally, i.e., without the need for assuming the steps are small. The machinery and details we plan to publish in a sequel article.

## 4. Relevance in Chemistry and Biology

Recently, Rubi et al. [14] analyzed the energetics and the kinetics of a general reversible chemical reaction passing through an intermediate state by comparing the rate and dissipation in the direct reaction A⇔C with the one with an intermediate state, A⇔B⇔C. In all situations, introducing the intermediate state B reduced the dissipation, although to a varying degree depending on the energy of the intermediate state. At the same time, the reaction rate decreased.

An illustrative biological example related to the above study is the electron transport chain in mitochondria [9]. The ladder theorem predicts that, if an additional step were to be added to the electron transport chain, for instance three intermediate molecules in the sequence, rather than the current two (ubiquinone and cytochrome-c), the result would be favorable in so far as it would reduce entropy production. Note, however, that, typically, work extraction in each of the four new steps will be smaller than in each of the previous three steps. That means that if the going unit of work is one molecule of ATP, the total free energy difference from start to end of the electron transport chain may not allow extraction of 4 ATP but only the current 3 ATP. The extra work extracted may be in the form of smaller units. However, the additional step may involve a significant energetic overhead for the cell in the form of producing and maintaining additional catalysts, thus providing a framework for defining cost–benefit analyses for the evolution of these systems.

Subsequent evolutionary processes can then select for the redox potential of the electron in the new molecule to be equally spaced between two existing redox states in the electron transport chain or, even better, select for equally spaced redox potentials of all three intermediate molecules. Although, by utilizing this theorem, one cannot predict which new molecule will arise in the electron transport chain, one can predict which molecule would provide the greatest fitness advantage and, thus, which molecule would most likely to be selected for and eventually fixed within a population. This theorem might also allow for thermodynamic-based evolutionary predictions at higher biological levels, such as where new organisms might benefit most in trophic chains or which new niche space would be the most-beneficial to occupy to receive the greatest bioenergetic benefit.

The ladder theorem shows that adding steps in an energy-extracting process will lessen the entropy production of the process, thereby allowing more work to be extracted from the available energy. Utilizing this theorem, we hypothesized that biological entities should be selected for metabolic pathways with a greater number of steps when energy is limited.

Haas et al., 2016, provided an example of this hypothesis in marine ecosystems, where they examined bacterial metagenomes from coral reefs in three different ocean basins (Atlantic, Pacific, and Indian Oceans) [15]. In these systems, the dominant benthic (ocean-floor-dwelling) primary producers determine the amount of available free energy to heterotrophic consumers by fixing energy from sunlight and turning it into bioavailable carbon via the process of photosynthesis. In areas where coral is the dominant primary producer, the bioavailable carbon is less reduced (i.e., more oxidized) and, thus, contains less free energy to drive heterotrophic metabolisms, whereas in areas where algae is the dominant primary producer, there is a surplus of free energy in the form of highly reduced bioavailable carbon [16]. Therefore, organisms in these coral-dominated systems are expected to be selected to minimize entropy production and maximize the yield of their metabolic processes. Thus, we predicted that organisms should be selected to use metabolic pathways with more steps in these energy-limited, coral-dominated environments.

This is indeed what is observed in the central carbon metabolism of the microbes in these ecosystems. The microbes in the coral-dominated, energy-limited environments have an enrichment in genes encoding for the Embden–Meyerhof–Parnas pathway [17], a central carbon catabolic pathway that has nine steps involved in the breakdown of sugars to pyruvate, whereas microbes in algal-dominated, energy-surplus environments had an enrichment for alternative central carbon metabolism pathways, such as the pentose phosphate pathway and the Entner–Doudoroff pathway, which contain fewer steps than the Embden–Meyerhof–Parnas pathway. Furthermore, there is a significant correlation between the amount of algae in a system and the ratio of the genes involved in the Embden–Meyerhof–Parnas pathway to genes involved in the Entner–Doudoroff pathway, as shown in Figure 4. Thus, we have evidence that there is indeed selection and subsequent change in gene frequencies for organisms to change the number of energy-extracting steps in a metabolic process depending on the scarcity of available free energy.

Given our more-steps-the-better rule, at least for efficiency, why do we not always increase the number of steps? The answer is because the marginal returns from an additional step have to be weighed in comparison to the costs in a cost–benefit analysis. Adding and maintaining additional enzymes for an extra step can be expensive.

## 5. Technical Examples

While most transport containers for liquid nitrogen (boiling point 77 K) are merely thermos bottles (an evacuated double-wall stainless steel container), containers for liquid helium (boiling point 4 K) are usually not evacuated, but instead, have hundreds of layers of thin film surrounding the inner container (Figure 5). These force thermal equilibration on each layer with only a small temperature difference with the next layer. This sharply reduces the total heat conduction from the surroundings and, thus, evaporation of helium compared to a single large temperature gap. The same principle is in operation when windows are constructed with several layers of glass in order to reduce heat loss.

Another step process where one can increase the thermal efficiency by increasing the number of equilibration steps is distillation. Each tray in the distillation column is a point of equilibrium between liquid and vapor. Every additional tray will, therefore, increase the column’s thermodynamic efficiency [7,8]. Isotope enrichment by whichever method used [19] follows the same principle: the more units, the more efficient it is.

Finally, let us mention photovoltaic conversion of sunlight to electricity. Traditional photovoltaic cells absorb light at and above a certain frequency, wasting anything below. One can add additional layers of converters adjusted to a wider range of frequencies and, thus, retain a larger fraction of the solar spectrum at its respective voltage [20]. This has been performed, but such cells are more expensive.

## 6. Thermodynamic Length in Quantum Physics

In recent years, thermodynamic length and its application to the design of minimally dissipating adiabatic processes in the slow-driving quantum regime has proven to be very successful [21,22,23,24,25,26,27,28]. Most of these applications involve continuous time processes, but step processes have occasionally been employed [27]. So far, however, we know of no application of our ladder theorem in this context; the step processes used always use optimal separation. There is considerable demand for further development of quantum thermodynamic devices, both for quantum computing and for nano-Kelvin refrigeration of complicated systems. This demand is sure to drive many new applications, and the ladder theorem is likely to play a role.

## 7. Conclusions

We proved the *ladder property* for any process consisting of steps: (i) if the distribution of steps is augmented by an additional step anywhere between two existing steps, the new sequence will be more efficient than the original one; (ii) for the best results, all the steps should next be reallocated to be equidistant in thermodynamic length. The ladder theorem emphasizes the importance of thermodynamic length as a central measure for dissipation in irreversible processes.

We also note that the superiority of many steps is well established empirically in, e.g., insulation and separation applications. We see the principle emerge in many biological and evolutionary situations. These results are closely related to the dissipation in sequential chemical processes as they appear in many, if not most biological systems, e.g.,: How is the direction of such a chain reaction ensured? How is specificity achieved? How are driven processes powered? Given our more-steps-the-better rule, at least for efficiency, why do we not always increase the number of steps? The answer is because the marginal returns from an additional step have to be weighed against the additional costs in a cost–benefit analysis. Finally, we expect the design of quantum devices, e.g., for quantum computing and for nano-Kelvin refrigeration of complicated systems, will benefit from utilizing the ladder property. We hope to return to those questions in a future publication.

## Figures and Tables

**Figure 1 entropy-25-00539-f001:**
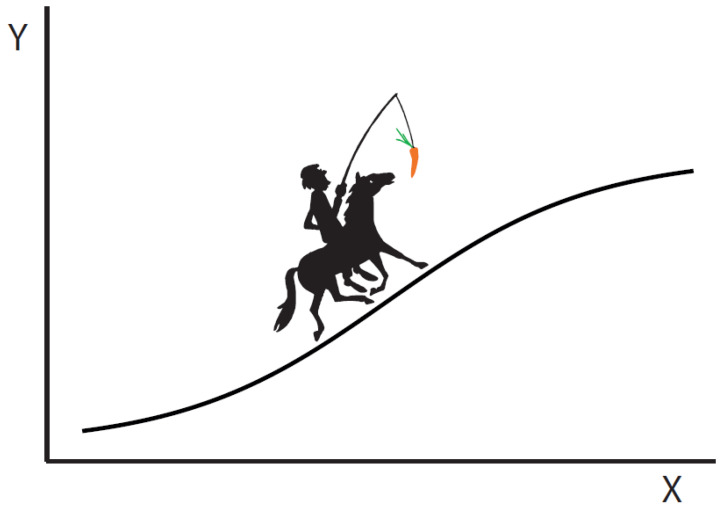
**A horse–carrot process.** The system (horse) keeps trying to relax to the position of the environment (carrot) along a prescribed path in the (Y,X) variables. In the thermodynamic example described below, those are pressure and volume (p,V). In a biological example, they might be, e.g., redox potential and charge (ions) moved (μ,N).

**Figure 2 entropy-25-00539-f002:**
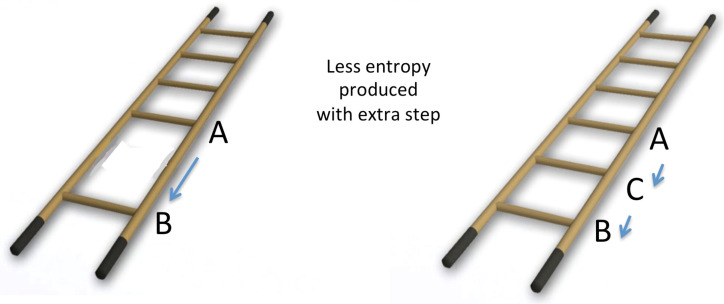
Inserting the step C decreases entropy production (see inequality in (Equation 7)).

**Figure 3 entropy-25-00539-f003:**
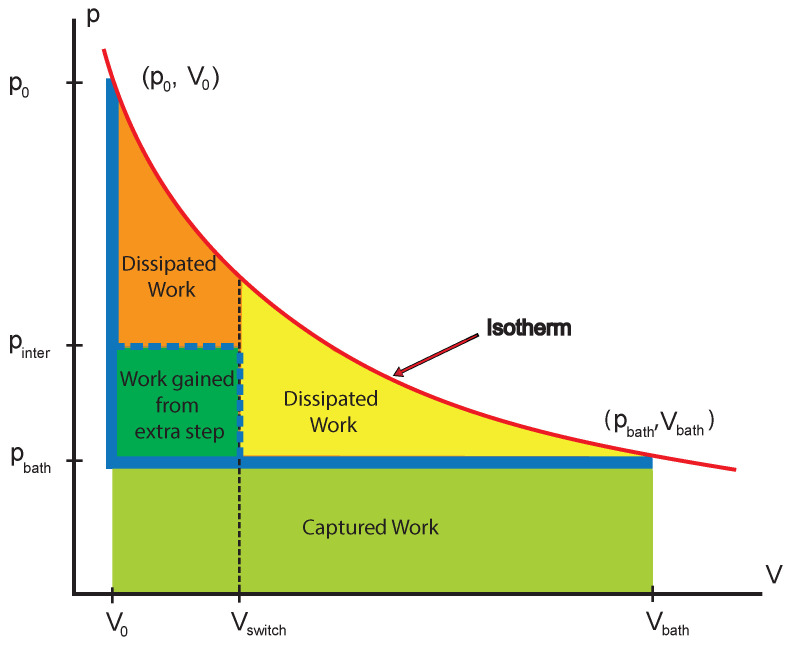
A gas is relaxing from $p_{0}$ to $p_{bath}$ along the isotherm of equilibrium points shown in red. Performing the relaxation in a single step from $p_{0}$ to $p_{bath}$ (blue line) dissipates the work corresponding to the area between the equilibrium curve and the actual path, i.e., the orange + yellow + dark-green areas. Introducing an additional step at $p_{inter}$ between $p_{0}$ and $p_{bath}$ and switching to it at $V_{switch}$ reduce the dissipation to the orange + yellow areas, thus increasing the work performed by the dark-green area. The light-green area below $p_{bath}$ also represents work performed by the system.

**Figure 4 entropy-25-00539-f004:**
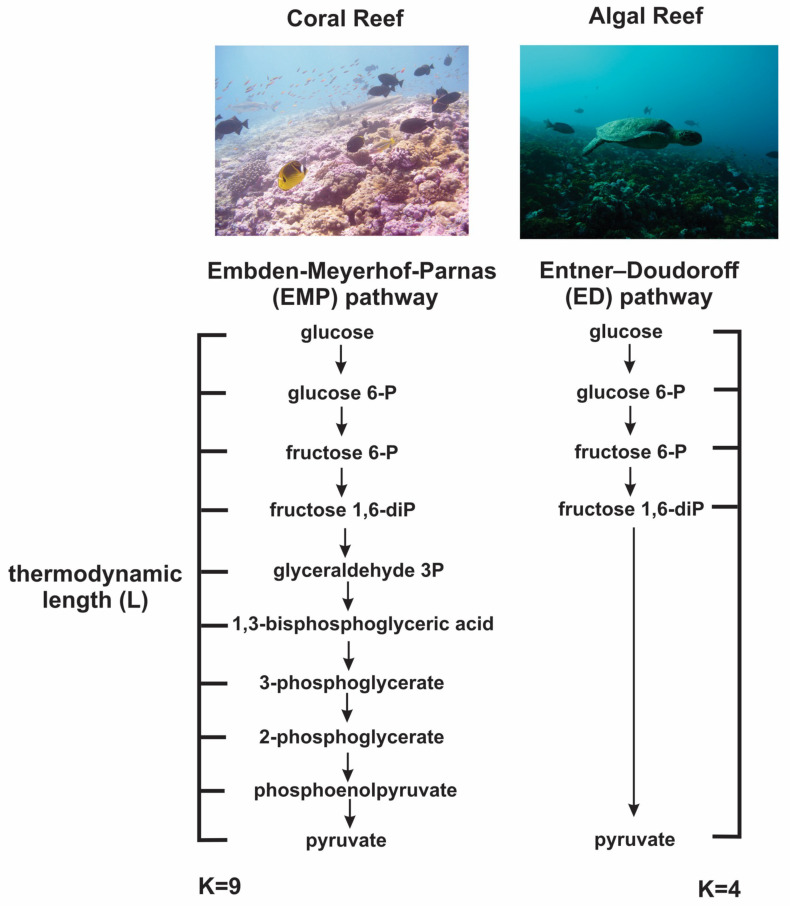
**The number of metabolic steps increases in environments with less bioavailable free energy.** Coral-dominated, energy-limited environments, such as those shown on the left, have a significant enrichment in genes encoding for the Embden–Meyerhof–Parnas pathway, a pathway that involves nine thermodynamic steps (K=9) in the breakdown of sugars to pyruvate, whereas algal-dominated, energy-surplus environments have an enrichment for alternative metabolic pathways, such as the Entner–Doudoroff pathway, which only contains four thermodynamic steps (K=4) [15]. Furthermore, areas on the reef that have an enrichment for metabolic pathways with fewer steps have also been shown to have lower thermodynamic efficiency in terms of heat released per cell [18], strengthening the link between thermodynamic length, step number, and efficiency in biological systems.

**Figure 5 entropy-25-00539-f005:**
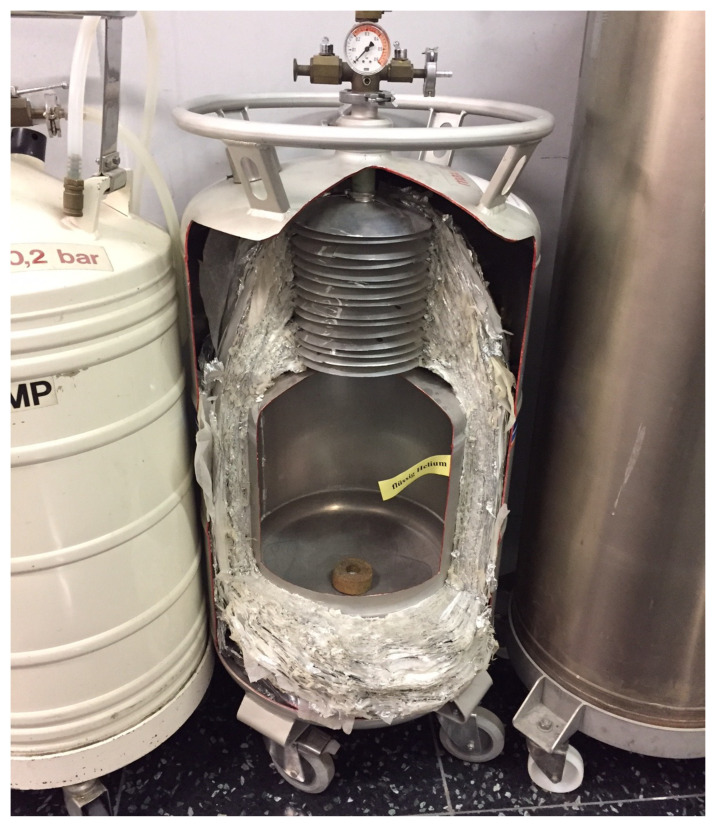
This liquid helium container minimizes the rate of heat leaking in by forcing literally hundreds of small equilibrations along the way.

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
