# Peer review of "More Stages Decrease Dissipation in Irreversible Step Processes"

_entropy, 2023, doi:10.3390/e25030539_

Round 1

Reviewer 1 Report

The paper is aimed at detailed thermodynamics consideration of the irreversible step process based on the thermodynamic length concept. As a result, the ladder theorem has been used for some particular cases. An advanced approach based on the either more steps in the process or shorter the thermodynamic length has been developed. The obtained results are very promising for applications.

Author Response

We thank the referee for his comments.

Reviewer 2 Report

PROCESS  IS A SEQUENCE OF SMALL EQUILBRATIONS

did it works for large desequilibrium ?

probably some limitations page 2 the time of the process goes to infinity p3

line v64 nearly endoreversible system      wu suggest to supress endo

page3 line103  expression (10)   where is it ? same line 108

page 4 define Xi

page 5 lines 160 161     yes   absolutely

exemples are illustrative

WHAT  IN TERM OF COST    PAGE 9 LINE 297   TO PRECISE AND EXPLORE

Author Response

PROCESS  IS A SEQUENCE OF SMALL EQUILBRATIONS

Correct

did it works for large disequilibrium ?

Yes, however we admit that this was the most important question not fully answered in the manuscript. Originally, we wrote it as a conjecture for larger steps. However, while the manuscript was out for review, we managed to prove. We have added a note to this effect at the end of section 3.

probably some limitations page 2 the time of the process goes to infinity p3

Sure, for infinite time the process becomes reversible. The power
expansion involved in the length calculation brings the process close to
equilibrium in each step as described by the ladder theorem.

line v64 nearly endoreversible system      would suggest to supress endo

Thank you for the suggestion, which we have adopted and changed the manuscript accordingly at the beginning of section 2.

page3 line103  expression (10)   where is it ? same line 108

Thank you for catching these references to equation (3) (formerly equation (10)) which had not been updated. We have corrected them.

page 4 define Xi

X refers to the set of independent extensive variables of the system. Thank you for pointing out the fact that we did not define the notation. This has now been corrected.

page 5 lines 160 161     yes   absolutely

We fear that the line numbers used by the referee do not match the numbering we have and there are not enough other clues concerning what the referee might have intended.

examples are illustrative

Thank you.

WHAT  IN TERM OF COST    PAGE 9 LINE 297   TO PRECISE AND EXPLORE

Great question but huge topic. To our knowledge, we are the first to worry about such cost-benefit analyses, or even to point to their importance. The slightly longer discussion (middle of section 4) is the best we can do for now.

We thank the referee for excellent comments that have certainly improved the manuscript.

Reviewer 3 Report

This manuscript is about the Ladder theorem, which states that a thermodynamic process gains efficiency when split into more steps, provided the steps proceed towards the equilibrium. 

The theorem is interesting and represents a general point of view on optimization. However, I think that the technical proof could be improved:

1) Formula A5 seems to be just the Taylor expansion to the second order around the equilibrium (where the first gradient disappears). Why do you need calculations A3-A4, then?

2) Formula A2 suggests that $\Delta S^u$ does not depend on the path, being a line integral of the exergy (thermodynamic potential, or available energy, see e.g. Landau & Lifshitz, volume 5), \Phi = S - \sum_i x^*_i x^i$, where $x^*_i$ are the equilibrium derivatives of entropy (equilibrium temperature, etc). The dependence on the path (number of steps, for instance), seems to be obtained only after the approximation of the path. 

3) Therefore, I suggest to reformulate A2 as a Riemannian sum approximating the line integral. This could also generalize the proof by omitting the second-order expansion limitations. The ladder theorem would then be the inequality between the line integral (represented by a Riemann integral) and the lower or upper (depends on the signs you choose) Riemann sums. 

In summary, I suggest a minor revision that could improve the technical details of the proof. 

Author Response

This manuscript is about the Ladder theorem, which states that a thermodynamic process gains efficiency when split into more steps, provided the steps proceed towards the equilibrium. 

Correct and nicely stated.

The theorem is interesting and represents a general point of view on optimization. However, I think that the technical proof could be improved:

1) Formula A5 seems to be just the Taylor expansion to the second order around the equilibrium (where the first gradient disappears). Why do you need calculations A3-A4, then?

Calculations A3-A4 are included to try and keep the exposition accessible to a less mathematically sophisticated audience. As the referee points out, they are not needed but merely insert steps for less capable calculators.

2) Formula A2 suggests that $\Delta S^u$ does not depend on the path, being a line integral of the exergy (thermodynamic potential, or available energy, see e.g. Landau & Lifshitz, volume 5), \Phi = S - \sum_i x^*_i x^i$, where $x^*_i$ are the equilibrium derivatives of entropy (equilibrium temperature, etc). The dependence on the path (number of steps, for instance), seems to be obtained only after the approximation of the path. 

While we are intrigued by the referee’s claim that the line integral in equation (A2) is path independent, we are not confident enough in his claims to change our entire approach here. Line integrals are generally path dependent. The referee’s claim is that this is exactly the line integral of the differential of exergy and thus its net change is just the change in exergy, regardless of the path. We agree that the suggested approach has promise but plan to explore it in a future effort.

3) Therefore, I suggest to reformulate A2 as a Riemannian sum approximating the line integral. This could also generalize the proof by omitting the second-order expansion limitations. The ladder theorem would then be the inequality between the line integral (represented by a Riemann integral) and the lower or upper (depends on the signs you choose) Riemann sums. 

In summary, I suggest a minor revision that could improve the technical details of the proof. 

We thank the referee for his suggestion and plan to explore it in a future effort.